# Effects of Two Physical Activity Interventions on Sleep and Sedentary Time in Pregnant Women

**DOI:** 10.3390/ijerph20075359

**Published:** 2023-03-31

**Authors:** Saud Abdulaziz Alomairah, Signe de Place Knudsen, Caroline Borup Roland, Stig Molsted, Tine D. Clausen, Jane M. Bendix, Ellen Løkkegaard, Andreas Kryger Jensen, Jakob Eg Larsen, Poul Jennum, Bente Stallknecht

**Affiliations:** 1Public Health Department, College of Health Sciences, Saudi Electronic University, Riyadh 13316, Saudi Arabia; 2Department of Biomedical Sciences, University of Copenhagen, 2200 Copenhagen, Denmark; 3Department of Clinical Medicine, University of Copenhagen, 2200 Copenhagen, Denmark; 4Department of Clinical Research, Copenhagen University Hospital—North Zealand, 3400 Hilleroed, Denmark; 5Department of Gynaecology and Obstetrics, Nordsjaellands Hospital, 3400 Hilleroed, Denmark; 6Department of Public Health, Section of Biostatistics, University of Copenhagen, 2200 Copenhagen, Denmark; 7Department of Applied Mathematics and Computer Science, Technical University of Denmark, 2800 Kongens Lyngby, Denmark; 8Danish Center for Sleep Medicine, Department of Clinical Neurophysiology, 2200 Copenhagen, Denmark

**Keywords:** consumer activity tracker, FitMum, maternal health, Pittsburgh Sleep Quality Index, Pregnancy Physical Activity Questionnaire, randomized controlled trial

## Abstract

Pregnancy is often associated with poor sleep and high sedentary time (SED). We investigated the effect of physical activity (PA) interventions on sleep and SED in pregnant women. A secondary analysis of a randomized controlled trial (*n* = 219) explored the effect of structured supervised exercise training (EXE) or motivational counseling on PA (MOT) compared to standard prenatal care (CON) on sleep and SED during pregnancy. Three times during pregnancy, sleep was determined by the Pittsburgh Sleep Quality Index (PSQI) and SED by the Pregnancy Physical Activity Questionnaire (PPAQ). Also, a wrist-worn consumer activity tracker measured sleep and SED continuously. Data from the activity tracker confirmed that sleep time decreases, and SED increases by approx. 30 and 24 min/day, respectively, from baseline (maximum gestational age (GA) week 15) to delivery. Compared to CON, the global PSQI score was better for EXE in GA week 28 (−0.8 [−1.5; −0.1], *p* = 0.031) and for both EXE and MOT in GA week 34 (−1 [−2; −0.5], *p* = 0.002; −1 [−2; −0.1], *p* = 0.026). In GA week 28, SED (h/day) from PPAQ was lower in EXE compared to both CON and MOT (−0.69 [−1; −0.0], *p* = 0.049; −0.6 [−1.0; −0.02], *p* = 0.042). In conclusion, PA interventions during pregnancy improved sleep quality and reduced SED.

## 1. Introduction

Pregnant women benefit from physical activity (PA) during pregnancy, including a decreased risk of excessive gestational weight gain, preterm birth, gestational diabetes mellitus, preeclampsia, delivery complications, and postpartum depression [1,2]. However, poor sleep quality during pregnancy might contradict the benefits [3,4]. Pregnancy-induced physiological and psychological changes include increased body weight, urination, anxiety, and stress [5]. Likewise, sleep is negatively affected, and sleep disturbances during pregnancy are more prevalent than in the general population [6]. One of the non-pharmaceutical ways to improve healthy sleep patterns in the general population is to engage in PA [7], and this is also true during pregnancy [8,9]. PA level is positively associated with sleep quality during pregnancy, and PA at both low and moderate intensity one to three days per week has been shown to improve sleep outcomes [8]. Yet, the evidence of which strategies of PA improve sleep during pregnancy is limited, and more robust randomized controlled trials (RCTs) that cover all trimesters are therefore needed [8,10].

Sedentary behavior is considered any physical behavior that does not significantly raise energy expenditure above that of resting (less than a 1.5 metabolic equivalence of the task), such as sleeping, sitting, lying down, watching television, and other screen-based activities [11]. The World Health Organization’s recommendations for pregnant women in 2020 replaced those issued in 2010 regarding PA and, for the first time, advised reducing the sedentary time (SED) [12]. A rising body of evidence suggests that SED may adversely affect adults’ health and be a risk factor for diabetes, cardiovascular disease, and death, independent of physical activity [13,14]. In addition, the prevalence of SED among pregnant women is higher than in the general population; pregnant women tend to spend more than 50% of their day as SED [15]. Thus, studies are needed to examine how interventions can effectively decrease SED while increasing PA levels during pregnancy.

Several PA interventions during pregnancy have focused on increasing moderate-to-vigorous-intensity PA (MVPA) and PA in general and evaluating the health effects of these interventions. However, few have focused on exploring the effect of PA interventions on sleep quantity and quality and SED during pregnancy [16]. We conducted a single-site three-armed RCT, the FitMum study, to evaluate the effects of offering structured supervised exercise training (EXE) or motivational counseling on PA (MOT) compared to standard prenatal care (CON) for inactive pregnant women [17]. Overall, we found that offering EXE was more effective than CON in increasing MVPA among pregnant women, whereas offering MOT was not [18]. The aim of the present secondary analysis was to assess the effect of the FitMum PA interventions on sleep quantity and quality and SED.

## 2. Methods

### 2.1. Ethics and Public Involvement

The FitMum study was approved by the Danish National Committee on Health Research Ethics (#H-18011067) and the Danish Data Protection Agency (#P-2019-512). The study adheres to the principles of the Helsinki declaration and is registered at ClinicalTrials.gov (NCT03679130). While designing the study, 27 semi-structured interviews with Danish pregnant women, midwives, and obstetricians were conducted. Before participants were included in the study, written informed consent was obtained.

### 2.2. Setting

This study was conducted at the Department of Gynaecology and Obstetrics at the public hospital Copenhagen University Hospital—North Zealand, Hillerød. Participation in the FitMum RCT was free of charge. The first participant was included in October 2018, and the last participant gave birth in May 2021.

### 2.3. Participants and Study Design

Two hundred twenty healthy pregnant women were included. Inclusion criteria were obtaining written informed consent, being 18 years or older, having a maximum gestational age (GA) of 15 weeks, having an ultrasonic-confirmed viable intrauterine pregnancy, having a body mass index of 18.5–45 kg/m^2^, and weighing <150 kg (pre-pregnancy weight or first measured weight in pregnancy), being able to wear a wrist-worn activity tracker 24/7 until delivery, and having a smartphone. Exclusion criteria were structured exercise at moderate-to-vigorous intensity for more than one hour per week during early pregnancy, previous preterm delivery, obstetric or medical complications, multiple pregnancies, non-Danish speaking, or alcohol or drug abuse.

### 2.4. Interventions

The aims and primary results of the FitMum study have been published elsewhere [17,18]. Briefly, we investigated 2 different strategies to increase PA in pregnant women with low PA levels and assessed the health effects of PA. The primary outcome was MVPA, measured by a Garmin Vivosport activity tracker. The FitMum RCT study had 3 study arms: (1) supervised structured exercise training (EXE), (2) motivational counseling on PA (MOT), and (3) standard prenatal care (CON). Participants in EXE and MOT were encouraged to be physically active at moderate intensity for at least 30 min daily. The EXE participants were offered 1-h supervised group sessions 3 times a week, 2 at the gym and 1 in the swimming pool. The MOT intervention consisted of weekly SMS reminders, 4 individual counseling sessions, and 3 group counseling sessions during pregnancy. Participants in all 3 study groups had 3 visits where sleep and SED were investigated: at baseline before GA week 15, at GA week 28, and at GA week 34. During the COVID-19 pandemic, restrictions in Denmark started on 11 March 2020 (total lockdown). The interventions shifted to online sessions and continued to be offered in that format until May 2021, when the intervention ended. The FitMum study had no intervention component regarding sleep or SED.

### 2.5. Outcomes

#### 2.5.1. Sleep Quantity and Quality by Pittsburgh Sleep Quality Index

The Danish version of the self-administered Pittsburgh Sleep Quality Index (PSQI) questionnaire [19,20] was digitally sent to the participants at baseline, GA week 28, and GA week 34. PSQI has been validated among pregnant women [21]. The PSQI has 19 questions that measure 7 components: (1) sleep quality, (2) sleep latency, (3) sleep duration, (4) sleep efficiency, (5) sleep disturbance, (6) use of sleep medication, and (7) daytime dysfunction. The sum of the 7 components forms the global PSQI score, ranging from 0 to 21, where a higher score indicates less sleep quality. A global PSQI score below 5 denotes a “good sleeper”, and a score above 5 indicates a “poor sleeper” [22].

#### 2.5.2. Sedentary Time by the Pregnancy Physical Activity Questionnaire

The Pregnancy Physical Activity Questionnaire (PPAQ) was designed and developed to determine PA intensity and duration during pregnancy [23]. We translated PPAQ to Danish and validated it in a Danish pregnant population [24]. For SED, we calculated time spent on sedentary activities from 5 questions as recommended [25,26] instead of 2 as done originally. Examples of sedentary behaviors assessed by PPAQ include “sitting at a desk during work or class” and “riding a car or bus”. PA duration and metabolic equivalence of task values were calculated according to the PPAQ developers’ guidelines; each answer in PPAQ corresponds to time spent in an activity multiplied by the intensity of the activity [27]. PPAQ was digitally sent to the participants at baseline, GA week 28, and GA week 34.

#### 2.5.3. Sleep and Sedentary Time by the Activity Tracker

The activity tracker data management and measurement details are published elsewhere [17,28]. In brief, all participants were given a consumer activity tracker with a built-in heart rate monitor and an accelerometer (Garmin Vivosport, Garmin, Olathe, KS, USA) [29], which had to be worn on the non-dominant wrist 24/7 from the inclusion until giving birth. Participants were instructed to sync the activity tracker data every day, and if a participant was not syncing for more than 7 days, an e-mail reminder would be sent. We monitored data flow and synchronization from the activity tracker through a research platform (Fitabase, San Diego, CA, USA). In contrast to the PSQI and PPAQ, the activity tracker determined sleep and SED continuously. The activity tracker combines heart rate and body movement data to determine when participants fall asleep, their awake time, and sleep stage during typical sleeping hours set by the user (not including nap time) [30]. We calculated sleep time as the sum of all sleep stages.

Moreover, the activity tracker shows PA daily values in a detailed log (Epoch log). From the Epoch log, a categorization of time is sorted into sedentary, active, or highly active by algorithms in the activity tracker. Sedentary is defined as little to no activity monitored; accordingly, minimal movement, sitting, resting, and sleeping are considered sedentary behavior [29]. We calculated SED by subtracting sleep time from total sedentary behavior. Data from the activity tracker was handled and included in the analysis according to predefined wear time criteria [28].

### 2.6. Statistical Analysis

For the PSQI and PPAQ outcomes, a constrained linear mixed model was fitted with the observation times as a factor [31], and the inference was performed based on a cluster bootstrap procedure. The between-group effects were reported as estimated differences in means. Intention-to-treat analyses using all randomized participants were performed for the outcomes from the activity tracker [28]. Missing observations in activity tracker data due to non-wear time was imputed by multiple imputations in 25 data sets using a pre-specified seed, pre-selected baseline variables (body weight, age, PA, educational level, sleep, SED, and parity), and the random forest imputation model from the mice R package [32]. For the activity tracker analysis, a constrained linear mixed model has been used of the mean values for baseline (6 days), randomization to GA week 28 (approx. 110 days), GA week 28 to GA week 34 (approx. 42 days), and GA week 34 to delivery (approx. 40 days), respectively. Sleep and SED before and during the COVID-19 pandemic were compared within groups with a linear regression model. All statistical analyses were performed using R version 4.2.2 [33]. Data are presented as means ± standard deviation for symmetric distributions and medians (interquartile ranges) for skewed data. The level of statistical significance was 5%, with 95% confidence intervals (CI) given for all reported estimates.

## 3. Results

### 3.1. Participant Characteristics

219 women were randomised to CON (*n* = 45), EXE (*n* = 87), or MOT (*n* = 87). At baseline, participants had a median GA of 12.9 weeks (9.4–13.9), age was 31.5 ± 4.3 years, and body weight was 75.4 ± 15.3 kg. The median pre-pregnancy body mass index was 24.1 (21.8–28.7) kg/m^2^. Participants wore the activity tracker for a total of 24,519 days out of 31,646 potential days (77%). The median activity tracker wear time was 183 (4–232) days. Lost to follow-up were 24% for CON, 15% for EXE and 20% for MOT from randomization to delivery. The adherence to intervention participation was 1.3 [95% CI 1.1; 1.5] exercise sessions per week from randomization to delivery for EXE, whereas MOT attended 5.2 [4.7; 5.7] counseling sessions from randomization to delivery.

### 3.2. Sleep Quantity and Quality by the Pittsburgh Sleep Quality Index

PSQI was completed by 219 (100%), 180 (82%), and 165 (75%) participants at baseline, GA week 28, and GA week 34, respectively. The mean global PSQI score (6.4 ± 1.9) was above 5 for all three groups at baseline. When comparing the two intervention groups with CON, EXE scored lower (i.e., lower means better) in the global PSQI score at GA week 28 (−0.8 [−2; −0.1], *p* = 0.031) and GA week 34 (−1 [−2; −0.5], *p* = 0.002; Table 1, Figure 1).

Also, MOT scored lower than CON at GA week 34 (−1 [−2; −0.1], *p* = 0.026; Figure 1, Table 1). There were no significant differences, except for sleep latency and sleep disturbance, when comparing EXE and MOT to CON for the individual PSQI outcomes (Table 1). At GA week 34, EXE had lower sleep latency (−0.5 [−0.8; 0.05], *p* = 0.027) and less sleep disturbance (−0.3 [−0.5; −0.05], *p* = 0.019) compared to CON. When comparing EXE to MOT, there were no significant differences for the individual PSQI outcomes, except that EXE scored lower than MOT for daytime dysfunction at GA week 28 (−0.2 [−0.4; −0.04], *p* = 0.017). A full comparison between the three groups is shown in Table 1. The average sleep time (h/day) decreased for all participants (time effect) from baseline to GA week 34 (−0.24 [−0.4; −0.1], *p* = 0.001; approx. 14 min/day).

### 3.3. Sedentary Time by the Pregnancy Physical Activity Questionnaire

At GA week 28, SED (h/day) from PPAQ was lower for EXE compared to both CON (−0.69 [−1; −0.0], *p* = 0.0498) and MOT (−0.6 [−1.0; −0.02], *p* = 0.042; Figure 2, Table 1).

Additionally, average SED (h/day) decreased among all participants (time effect) from baseline to GA week 34 (−1.1 [−1.5; −0.67], *p* < 0.001; Figure 2).

### 3.4. Sleep and Sedentary Time by the Activity Tracker

The unadjusted average of sleep time (h/day) for all participants was (8.2 [8.1; 8.3]), (8.0 [7.9; 8.1]) and (7.8 [7.8; 7.9]), respectively, at GA week 28, GA week 34 and delivery. Moreover, the unadjusted average SED (h/day) for all participants was (13.1 [12.9; 13.2]), (13.2 [13.0; 13.3]) and (13.5 [13.2; 13.6]), respectively, at GA week 28, GA week 34 and delivery. However, sleep time and SED did not differ significantly between groups (Table 2 and Figure 3).

Compared to the baseline, the average sleep time (h/day) decreased for all participants at GA week 28 (−0.2 [−0.3; −0.1], *p* <.001), GA week 34 (−0.4 [−0.4; −0.2], *p* < 0.001), and delivery (−0.5 [−0.6; −0.4], *p* < 0.001; approx. 12, 18 and 30 min/day, respectively). On the other hand, the average SED (h/day) increased among the participants as the pregnancy progressed and was significantly higher at delivery compared with baseline (0.4 [0.2; 0.5], *p*< 0.001; approx. 24 min/day).

### 3.5. COVID-19 Impact on Sleep and Sedentary Time as Measured by the Activity Tracker

No overall differences in sleep time and SED from randomization to delivery were found between participants ending the intervention before the COVID-19 pandemic (physical intervention only, *n* = 120) and those included and ending the intervention during the COVID-19 pandemic (online intervention only, *n* = 63). However, EXE participants who were offered the online intervention during the COVID-19 pandemic had more SED (h/day) than those offered the physical intervention (0.4 [−0.1; 0.8], *p* = 0.032; approx. 25 min/day; Figure 4).

### 3.6. Comparison of Sleep Time from the Activity Tracker and the Pittsburgh Sleep Quality Index

We compared sleep time from the activity tracker and PSQI. At baseline, GA week 28, and GA week 34, the correlations were weak (r = 0.17, 0.27, and 0.31 (*p* = 0.01, 0.001 and 0.001), respectively). The mean biases for sleep time between the activity tracker and PSQI were 1.2, 1.0, and 1.0 h/day, respectively, with higher values reported by the activity tracker than by the PSQI (Figure 5).

## 4. Discussion

In this secondary analysis of the FitMum RCT, we found that the overall sleep quality, as determined by PSQI, was better in EXE than CON at GA week 28 and better in both EXE and MOT than CON at GA week 34. Moreover, EXE had less SED than MOT and CON at GA week 28, according to PPAQ. The activity tracker showed no significant differences between groups in sleep time and SED. However, sleep time decreased as the pregnancy progressed. SED constituted more than half of the day and increased toward the end of the pregnancy. Moreover, participants in EXE who received the intervention online due to COVID-19 restrictions had more SED than those who received the physical EXE intervention before COVID-19.

### 4.1. Effectiveness of Physical Activity Interventions on Sleep Quality as Determined by the Pittsburgh Sleep Quality Index

We observed a relatively high mean global PSQI score at baseline, which is similar to other findings among pregnant women [6,34,35]. In alignment with our results, a recent systematic review showed that PA level was positively associated with sleep quality as determined by the PSQI during pregnancy [8]. In addition, a systematic review and meta-analysis of RCTs conducted among pregnant women revealed that sleep quality was improved among exercise group participants when determined by the PSQI [36]. Like our findings, an RCT of an 8-week supervised home tele-based Pilates program 50 min twice a week (*n* = 7) and control (*n* = 7) during pregnancy showed that PSQI global scores were significantly lower in the intervention compared to the control group [37]. In contrast to our findings, an RCT among Danish pregnant women with or at high risk of depression found no difference in the global PSQI score after 12 weeks of supervised group exercise (70 min twice a week) starting from GA 17–22 weeks [35]. However, women participating in >74% of the exercise sessions (per protocol analysis) had significantly lower mean global PSQI scores than women in the control group.

### 4.2. Effectiveness of Physical Activity Interventions on Sedentary Time as Determined by the Pregnancy Physical Activity Questionnaire

The PPAQ showed lower SED measured in EXE compared to MOT and CON, which contradicts other findings. A 12-week unsupervised exercise intervention in early pregnancy did not affect SED [38]. Moreover, pregnant women randomized to 12 weeks of supervised exercise three times a week spent more time performing MVPA than the control group, but SED reported by PPAQ did not differ between groups [39]. Also, 90 pregnant women were randomized to an 8-week educational intervention on WhatsApp to improve PA or to a control group. PA level was increased in the intervention group, but SED measured by PPAQ did not differ between groups [40]. These contradictory results might be because we used five instead of two items from PPAQ to compute SED as recently recommended [25,26]. In this way, we increased the sensitivity of the questionnaire.

### 4.3. Sleep and Sedentary Time as Determined by the Activity Tracker

Like others, we found that sleep decreases [3,4,5], and SED may increase as pregnancy progresses [16,41]. Notably, device-based methods have not been used in previous RCT studies investigating the effects of PA on sleep among pregnant women [8,36]. A systematic review investigating sedentary behaviors during pregnancy found that despite the wide disparity between sedentary behavior definitions and measurement techniques, pregnant women spent more than half of their day in SED [15], which aligns with our findings. In addition, few studies examined sleep using a consumer activity tracker during most of the pregnancy period. An observational study that used Fitbit Flex to examine pregnant women’s sleep duration discovered a strong inverse correlation between sleep and GA [42].

### 4.4. Sedentary Time Measured by the Pregnancy Physical Activity Questionnaire and the Activity Tracker

We observed that SED increased during pregnancy when measured by the activity tracker (approx. 24 min/day) and decreased when measured by PPAQ (approx. 1 h/day). This might be explained by two previous findings from the FitMum study. First, in a validation study, we found a significant underestimation of SED by PPAQ compared to the activity tracker [43]. The mean biases were 6.8, 7.2 and 8.1 h/day, respectively, at baseline, GA week 28 and GA week 34. Hence, distinct PA constructs are determined by PPAQ and the activity tracker. Secondly, the PA dose in EXE was delivered with high fidelity [44]. This could influence EXE participants’ perception of SED, thus reporting less SED in the PPAQ. Although combining various methods to measure SED during pregnancy gives a comprehensive assessment, rigorous studies are needed to gain better knowledge about SED during pregnancy.

### 4.5. Validity of Activity Trackers for Measuring Total Sleep Time

The validity of the Garmin Vivosport in measuring sleep during pregnancy has not been tested before. One study reviewed the validity of Garmin activity trackers, not including Garmin Vivosport, in measuring sleep and found that sleep time was overestimated by the activity trackers when using a sleep diary as a criterion method [45], which is in alignment with our data. Also, when using polysomnography as a criterion method, other brands of activity trackers tend to overestimate sleep time and underestimate wake time after sleep onset [45,46]. A recent study investigated the validity of three consumer activity trackers, including Garmin Vivosport, in older adults and found that all three activity trackers had a high level of accuracy for measuring sleep time [47].

### 4.6. Strengths and Limitations

PA, sleep and SED during pregnancy are difficult to evaluate accurately. It is a strength that this study utilized both reported and device-based methods at different times during pregnancy. The activity tracker was advantageous to continuously capturing sleep time and SED throughout pregnancy. However, the consumer activity tracker’s validity, adaptability, and applicability in research and clinical practice need standardization and consensus [48]. The manufacturer processes the sleep and SED measures from the activity tracker, and the algorithms have not been published; for instance, how the activity tracker distinguishes between sitting, lying, and standing, or sleep stages, is proprietary information. Also, self-reported and device-based measures of sleep and SED may assess distinct constructs, resulting in weak correlations between the activity tracker and questionnaires. Moreover, the inherited bias from self-reported sleep and SED from the questionnaires is inevitable. An additional limitation of this study is that the analysis is secondary; hence, no sample size nor power calculation was made on the outcomes of the present analyses.

## 5. Conclusions

This study affirmed that pregnant women are prone to low sleep quality and high SED, which worsens as pregnancy progresses. Pregnant women who received structured supervised exercise training had better sleep quality and less SED than pregnant women receiving standard prenatal care when self-reported. When measured by the consumer activity tracker, no differences were observed between groups. In an online setting, due to COVID-19 restrictions, SED was increased among pregnant women who received the EXE intervention. In conclusion, interventions that increase PA levels might improve sleep quality and decrease SED in pregnant women. Future behavioral interventions targeting pregnant women should include evidence-based content to improve sleep quality and reduce SED.

## Figures and Tables

**Figure 1 ijerph-20-05359-f001:**
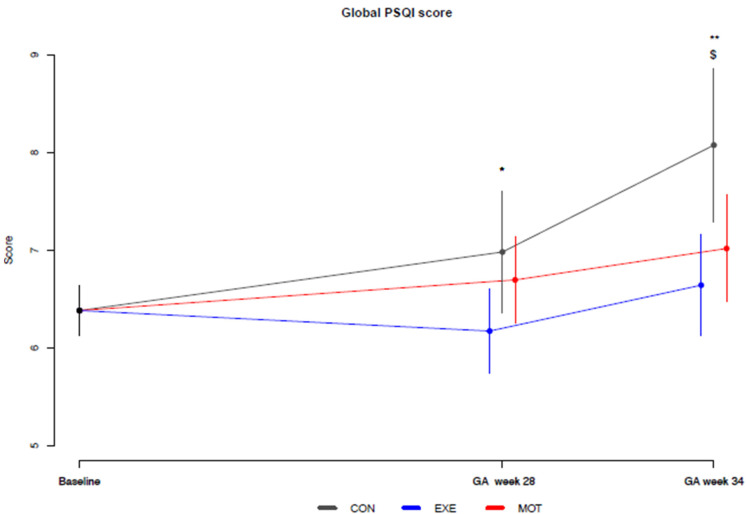
Baseline-constrained comparison between groups based on the means of the Pittsburgh Sleep Quality Index global score. Baseline, gestational age of maximum 15 weeks; CON (gray color), standard care; EXE (blue color), structured supervised exercise training; GA, gestational age; MOT (red color), motivational counseling on physical activity. * EXE compared to CON at GA week 28 (*p* = 0.031), ** EXE compared to CON at GA week 34 (*p* = 0.002), $ MOT compared to CON at GA week 34 (*p* = 0.026).

**Figure 2 ijerph-20-05359-f002:**
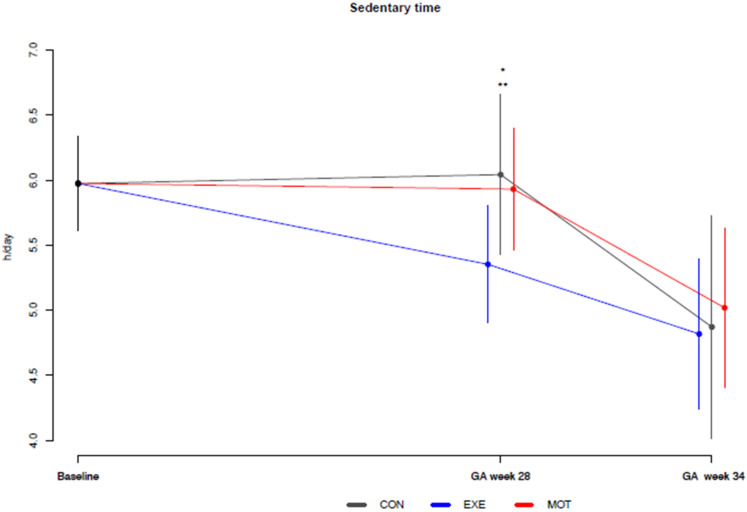
Baseline-constrained comparison between groups based on the means of sedentary time from the Pregnancy Physical Activity Questionnaire. Baseline, gestational age of maximum 15 weeks; CON (grey color), standard care; EXE (blue color), structured supervised exercise training; GA, gestational age; hr, hour; MOT (red color), motivational counseling on physical activity. * EXE compared to CON at GA week 28 (*p* = 0.0498), ** EXE compared to MOT at GA week 28 (*p* = 0.042).

**Figure 3 ijerph-20-05359-f003:**
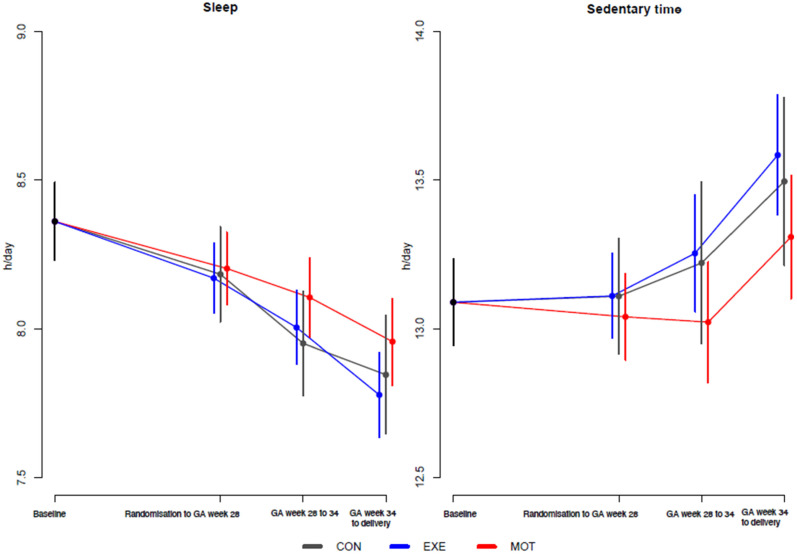
Baseline-constrained comparison between groups based on the activity tracker’s mean of total sleep and sedentary time. Baseline, gestational age of maximum 15 weeks; CON (gray color), standard care; EXE (blue color), structured supervised exercise training; GA, gestational age. h, hour; MOT (red color), motivational counseling on physical activity.

**Figure 4 ijerph-20-05359-f004:**
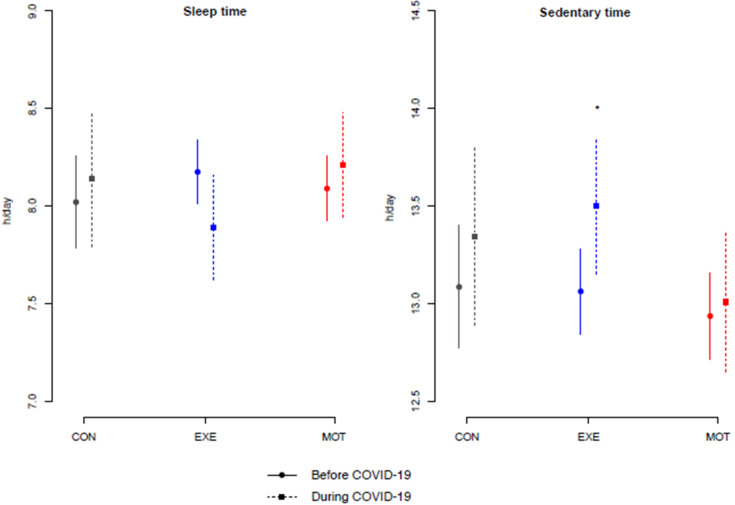
Average and 95% confidence interval of total sleep time and sedentary time before COVID-19 [physical intervention only, participants (*n* = 120) started and finished the intervention before COVID-19] (full line) and during COVID-19 [online intervention only, participants (*n* = 63)] (dotted line)], respectively. CON (gray color), standard care; EXE (blue color), structured, supervised exercise training; h, hour; MOT (red color), motivational counseling on physical activity. * EXE who received the physical intervention compared to EXE who received online intervention from randomization (gestational age of maximum 15 weeks) to delivery (*p* = 0.032).

**Figure 5 ijerph-20-05359-f005:**
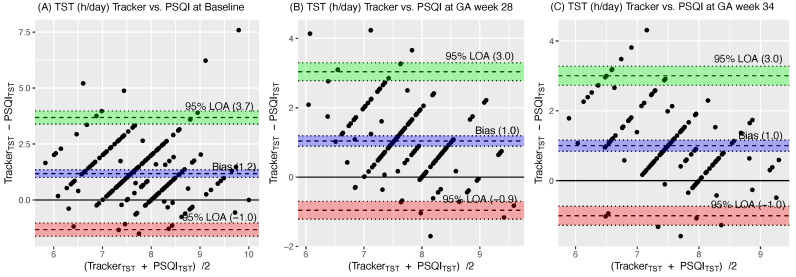
Differences in the total sleep time between the Pittsburgh Sleep Quality Index (PSQI) and the activity tracker vs. the average of sums at the gestational age of maximum 15 weeks (**A**), GA week 28 (**B**), GA week 34 s (**C**). h, hour; GA, gestational age; TST, total sleep time; LOA, the limit of agreements.

**Table 1 ijerph-20-05359-t001:** Outcomes from the Pittsburgh Sleep Quality Index and sedentary time from the Pregnancy Physical Activity Questionnaire.

	CON vs. EXE				CON vs. MOT				MOT vs. EXE			
	GA Week 28		GA Week 34		GA Week 28		GA Week 34		GA Week 28		GA Week 34	
	Differences [95% CI]	*p*-Value	Differences [95% CI]	*p*-Value	Differences [95% CI]	*p*-Value	Differences [95% CI]	*p*-Value	Differences [95% CI]	*p*-Value	Differences [95% CI]	*p*-Value
PSQI												
Global PSQI score	−0.8 [−2; −0.1]	**0.031**	−1 [−2; −0.5]	**0.002**	−0.3 [−1.0; 0.5]	0.451	−1.0 [−2; −0.1]	**0.026**	−0.5 [−0.1.1; 0.1]	0.848	−0.4 [−1.1; 0.3]	0.309
Total sleep time (h/day)	0.06 [−0.3; 0.41]	0.727	0.1 [−0.3; 0.5]	0.702	0.11 [−0.24; 0.5]	0.554	0.27 [−0.14; 0.7]	0.191	−0.04 [−0.3; 0.2]	0.757	−0.2 [−0.5; 0.1]	0.234
Total time in bed (h/day)	−0.20 [−0.6; 0.17]	0.282	−0.1 [−0.5; 0.3]	0.637	−0.14 [−0.5; 0.2]	0.474	0.02 [−0.4; 0.4]	0.921	−0.1 [−0.4; 0.2]	0.686	−0.1 [−0.4; 0.2]	0.469
Subjective sleep quality	0.11 [ 0.1; 0.3]	0.340	0.03 [−0.24; 0.32]	0.795	0.11 [−0.12; 0.35]	0.341	0.1 [−0.2; 0.4]	0.494	−0.005 [−0.2; 0.2]	0.961	−0.05 [−0.30; 0.2]	0.661
Sleep efficiency (%)	3 [−0.9; 7]	0.133	3 [−2; 8]	0.240	2.7 [−1.4; 6.9]	0.199	3.7 [−1.3; 8.86]	0.146	0.4 [−2.9; 3.8]	0.804	−0.8 [−4.7; 3.1]	0.696
Sleep Disturbance	−0.14 [−0.3; 0.1]	0.164	−0.3 [−0.5; −0.05]	**0.019**	−0.07 [−0.3; 0.1]	0.485	−0.14 [−0.4; 0.11]	0.279	−0.1 [−0.24; 0.1]	0.388	−0.2 [−0.3; 0.03]	0.098
Sleep Medications	−0.1 [−0.3; 0.07]	0.246	−0.1 [−0.3; 0.1]	0.324	0.0003 [−0.2; 0.2]	0.997	−0.02 [−0.2; 0.15]	0.801	−0.1 [−0.2; 0.04]	0.153	−0.1 [−0.2; 0.1]	0.344
Sleep latency	−0.33 [−0.7; 0.04]	0.077	−0.5 [−0.8; 0.05]	**0.027**	−0.2 [−0.6; 0.1]	0.221	−0.21 [−0.6; 0.2]	0.325	−0.09 [0.4; −0.2]	0.541	−0.25 [−0.5; 0.04]	0.098
Daytime Dysfunction	−0.03 [−0.3; 0.2]	0.758	−0.3 [−0.5; 0.00]	0.052	0.20 [−0.03; 0.4]	0.095	−0.146 [−0.43; 0.14]	0.313	−0.2 [−0.4; −0.04]	**0.017**	−0.13 [−0.32; 0.1]	0.186
PPAQ												
Sedentary time h/day	−0.69 [−1; −0.0]	**0.0498**	−0.05 [−1; –0.9]	0.916	−0.11 [−0.8; −0.6]	0.754	0.1 [−0.8; −1.2]	0.776	−0.6 [−1.0; −0.02]	**0.042**	−0.1 [−1.0; 0.6]	0.776

Comparison between groups on sleep outcomes from Pittsburgh Sleep Quality Index (PSQI) and sedentary time from the Pregnancy Physical Activity Questionnaire (PPAQ). A positive mean value indicates that the last-mentioned group has the highest mean. Bold *p*-value denotes statistical significance. CI, confidence interval; CON, standard care; EXE, structured supervised exercise training; GA, gestational age; h, hour; MOT, motivational counseling on physical activity.

**Table 2 ijerph-20-05359-t002:** Sleep and sedentary time from an activity tracker.

	CON vs. EXE						CON vs. MOT						MOT vs. EXE					
	GA Week 28		GA Week 34		Delivery		GA Week 28		GA Week 34		Delivery		GA Week 28		GA Week 34		Delivery	
	Differences [95% CI]	*p*-Value	Differences [95% CI]	*p*-Value	Differences [95% CI]	*p*-Value	Differences [95% CI]	*p*-Value	Differences [95% CI]	*p*-Value	Differences [95% CI]	*p*-Value	Differences [95% CI]	*p*-Value	Differences [95% CI]	*p*-Value	Differences [95% CI]	*p*-Value
Tracker outcomes																		
Total sleep time (h/day)	−0.01 [−0.2; 0.2]	0.890	0.05 [−0.1; 0.3]	0.603	−0.07 [−0.3; 0.2]	0.573	0.02 [−0.2; 0.2]	0.833	0.15 [−0.1; 0.4]	0.151	0.1 [−0.1 to 0.4]	0.366	−0.03 [−0.2; 0.1]	0.669	−0.1 [−0.3; 0.1]	0.247	−0.08 [−0.4; 0.02	0.072
Sedentary time (h/day)	0.001 [−0.2; 0.2]	0.993	0.03 [−0.3; 0.4]	0.845	0.1 [−0.3; 0.4]	0.609	−0.1 [−0.3; 0.1]	0.534	−0.2 [−0.5; 0.1]	0.234	−0.1 [−0.5; 0.2]	0.285	0.1 [−0.1; 0.2]	0.437	0.2 [−0.04; 0.5]	0.095	0.3 [−0.03; 0.5]	0.053

Comparison between groups based on imputed activity tracker datasets (intention to treat analysis) from randomization (gestational age of maximum 15 weeks), GA week 28, GA week 34 and delivery, respectively. A positive mean value indicates that the last-mentioned group has the highest mean. CI, confidence interval; CON, standard care; EXE, structured supervised exercise training; GA, gestational age; h, hour; MOT, motivational counseling on physical activity.

## Data Availability

Due to confidentiality, the datasets utilized in the current work are not publicly accessible. However, data are available upon justifiable request from the corresponding author, obtained the Danish Data Protection Authority’s consent per the Data Protection Act, and finished a Standard Contractual Clause to ensure the transfer’s legal basis.

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
