# Peer review of "Effects of Two Physical Activity Interventions on Sleep and Sedentary Time in Pregnant Women"

_ijerph, 2023, doi:10.3390/ijerph20075359_

Round 1

Reviewer 1 Report

Effects of two physical activity interventions on sleep and sedentary time in pregnant women

This is a novel study conducted within the context of a wider randomised controlled trial, investigating the effects to two distinct physical activity interventions on self-reported and device-assessed measures of sleep and sedentary time in pregnant women. Overall, the manuscript is well written, although I have identified a few areas requiring further clarification. While the methods employed are robust, I suggest incorporating further analysis to examine responsiveness to change in both device-based and self-report measures of sleep and sedentary time between the distinct physical activity interventions and time points. I hope my comments and suggestions outlined below are useful to the authors.

Introduction

Page 2, lines 55-57: You said: “A rising body of evidence suggests that SED may adversely affect adults' health and be a risk factor for diabetes, cardiovascular disease, and death”. I suggest adding further clarity regarding the potentially detrimental influence of sedentary behaviour on health, independent of physical activity (i.e. whether or not an individual meets levels of physical activity sufficient for health benefits). For example: “A rising body of evidence suggests that SED may adversely affect adults' health and be a risk factor for diabetes, cardiovascular disease, and death, independent of physical activity.”

Page 2, lines 57-59: It is important to convey that pregnant women may not be protected from the deleterious influence of sedentary behaviour on health even if they are physical active (e.g. 30 minutes per day). Hence, it is important to make clear that there is a need to focus on reducing sedentary time in combination with increasing physical activity among pregnant women.  

Methods

Page 3, lines 107-109: There were differences in the level and type of restrictions imposed by difference governments during COVID-19, particularly at the beginning of the pandemic (Spring, 2020). While you stipulate those restrictions in Denmark started on March 11, you do not specify the level of restrictions imposed (e.g. total lockdown) and for how long these measures were imposed. This information would provide useful context. Moreover, it would be enlightening to know for how long the intervention sessions shifted to online sessions and if the impact of this had any ramifications (e.g. access to intervention content across sub-groups etc).

Page 3, lines 123-125: It would be useful to see examples of the questions used from the PPAQ to assess self-reported sedentary time.

Results

It has been suggested that although related, device-based, and self-report measures assess distinct physical activity constructs, thus may not be comparable (see Fulton et al., 2016) https://journals.lww.com/acsm-msse/Fulltext/2016/10000/Strategic_Priorities_for_Physical_Activity.24.aspx). Hence, it could be argued that self-report and device-based measures of sedentary time and sleep measure distinct constructs, thus it is not surprising that the correlations between the activity tracker and PSQI for sleep were weak. It would therefore be enlightening to examine responsiveness to change in device-based and self-report measures of sedentary time and sleep scores between the distinct physical activity interventions and time points, for instance, using the Standardised Response Mean (SRM) and Guyatt Responsiveness Index (GRI) (see Husted et al, 2000) https://www.sciencedirect.com/science/article/pii/S0895435699002061?via%3Dihub

Discussion

Page 10, lines 336: “PA behaviour during pregnancy is difficult to evaluate accurately”: It is worth adding that sedentary time and sleep are also difficult to evaluate as although they are related to PA they are in fact distinct constructs.  

Page 10, lines 336-336: “It is a strength that 336 this study utilised both subjective and objective methods at different times during pregnancy”: while this is the case, I suggest revising the use of constructs both here and consistently throughout the rest of the manuscript to reflect current conventions, namely ‘reported’ and ‘device-based’. For instance, Fulton et al (2016), page 2067, stipulate:  

“Accurate terminology is also important for clear communication. When terminology does not accurately represent a situation or construct, it may produce a bias in use or interpretation. For example, it is common to refer to reported or questionnaire-based assessments of physical activity as “subjective” and to device-based assessments as “objective.” These terms suggest that one method is superior to the other. To reduce the potential for bias, roundtable experts recommended using the terms “reported” and “device-based” instead. Those who oversee publishing for research and practice audiences may wish to adopt similar terminology standards.”

Conclusion

Page 11, lines 348-355: It is important to highlight that in addition to increasing physical activity, future behavioural interventions targeting pregnant women, should incorporate evidence-based content explicitly aimed at reducing sedentary time and enhancing sleep quality.  

Author Response

Response to Reviewer 1 Comments

Point 1:

Page 2, lines 55-57: You said: “A rising body of evidence suggests that SED may adversely affect adults' health and be a risk factor for diabetes, cardiovascular disease, and death”. I suggest adding further clarity regarding the potentially detrimental influence of sedentary behaviour on health, independent of physical activity (i.e. whether or not an individual meets levels of physical activity sufficient for health benefits). For example: “A rising body of evidence suggests that SED may adversely affect adults' health and be a risk factor for diabetes, cardiovascular disease, and death, independent of physical activity.”

Response 1: We agree with the point and added “independent of physical activity” to the sentence (line 56).

Point 2: Page 2, lines 57-59: It is important to convey that pregnant women may not be protected from the deleterious influence of sedentary behaviour on health even if they are physical active (e.g. 30 minutes per day). Hence, it is important to make clear that there is a need to focus on reducing sedentary time in combination with increasing physical activity among pregnant women. 

Response 2: We now clarify the sentence: “Thus, studies are needed to examine how interventions can effectively decrease SED while increasing PA levels during pregnancy” (lines 58-59).

Point 3: Page 3, lines 107-109: There were differences in the level and type of restrictions imposed by difference governments during COVID-19, particularly at the beginning of the pandemic (Spring, 2020). While you stipulate those restrictions in Denmark started on March 11, you do not specify the level of restrictions imposed (e.g. total lockdown) and for how long these measures were imposed. This information would provide useful context. Moreover, it would be enlightening to know for how long the intervention sessions shifted to online sessions and if the impact of this had any ramifications (e.g. access to intervention content across sub-groups etc).

Response 3: Thank you for drawing our attention to this point. We added what type of restrictions was imposed at the beginning of the pandemic “(total lockdown).” and that the intervention shifted online “and continued to be offered in that format until May 2021, when the intervention ended” (lines 110 and 111).

Point 4: Page 3, lines 123-125: It would be useful to see examples of the questions used from the PPAQ to assess self-reported sedentary time.

Response 4: We added a few examples of sedentary behaviours assessed by PPAQ: “sitting at a desk during work or class” and “riding a car or bus”. (lines 128 and 129).

Point 5: It has been suggested that although related, device-based, and self-report measures assess distinct physical activity constructs, thus may not be comparable (see Fulton et al., 2016) https://journals.lww.com/acsm-msse/Fulltext/2016/10000/Strategic_Priorities_for_Physical_Activity.24.aspx). Hence, it could be argued that self-report and device-based measures of sedentary time and sleep measure distinct constructs, thus it is not surprising that the correlations between the activity tracker and PSQI for sleep were weak. It would therefore be enlightening to examine responsiveness to change in device-based and self-report measures of sedentary time and sleep scores between the distinct physical activity interventions and time points, for instance, using the Standardised Response Mean (SRM) and Guyatt Responsiveness Index (GRI) (see Husted et al, 2000) https://www.sciencedirect.com/science/article/pii/S0895435699002061?via%3Dihub

Response 5: While we acknowledge this point, we would like to clarify that our paper aimed not to compare self-report and device-based measures nor to test the validity or reliability of the activity tracker. Rather, our focus was on evaluating the effectiveness of physical activity interventions on sedentary time and sleep outcomes, using a combination of self-report and device-based measures. Additionally, we found it relevant to report the agreement between the activity tracker and PSQI. We now acknowledged this issue in the section on limitations: “Also, self-reported and device-based measures of sleep and SED may assess distinct constructs, resulting in weak correlations between the activity tracker and questionnaires.” (lines 352-354). Additionally, we recently published a validation paper (Alomairah et al, Methods to Estimate Energy Expenditure, Physical Activity, and Sedentary Time in Pregnant Women: A Validation Study Using Doubly Labeled Water, Journal for the Measurement of Physical Behaviour 2023) in which we aimed to compare the consumer activity tracker and the Danish version of PPAQ (PPAQ-DK) and to validate them using the doubly labeled water technique as criterion method.  In the current manuscript we refer to this paper and suggest the distinct constructs as a reason for inconsistency in SED results; “First, in a validation study, we found a significant underestimation of SED by PPAQ compared to the activity tracker [43]. The mean biases were 6.8, 7.2 and 8.1 hr/day, respectively, at baseline, GA week 28 and GA week 34. Hence, distinct PA constructs are determined by PPAQ and the activity tracker.” (lines 324-327).

Point 6: Page 10, lines 336: “PA behaviour during pregnancy is difficult to evaluate accurately”: It is worth adding that sedentary time and sleep are also difficult to evaluate as although they are related to PA they are in fact distinct constructs.

Response 6: We acknowledge your attention to this point and have edited it to “PA, sleep and SED during pregnancy are difficult to evaluate accurately” (line 344).

Point 7: Page 10, lines 336-336: “It is a strength that this study utilised both subjective and objective methods at different times during pregnancy”: while this is the case, I suggest revising the use of constructs both here and consistently throughout the rest of the manuscript to reflect current conventions, namely ‘reported’ and ‘device-based’. For instance, Fulton et al (2016), page 2067, stipulate: 

“Accurate terminology is also important for clear communication. When terminology does not accurately represent a situation or construct, it may produce a bias in use or interpretation. For example, it is common to refer to reported or questionnaire-based assessments of physical activity as “subjective” and to device-based assessments as “objective.” These terms suggest that one method is superior to the other. To reduce the potential for bias, roundtable experts recommended using the terms “reported” and “device-based” instead. Those who oversee publishing for research and practice audiences may wish to adopt similar terminology standards.”

Response 7: We appreciate this notation and information. We now addressed and changed these terms in the manuscript accordingly (lines 311, 345, 352, 354 and 374).

Point 8: Page 11, lines 348-355: It is important to highlight that in addition to increasing physical activity, future behavioural interventions targeting pregnant women, should incorporate evidence-based content explicitly aimed at reducing sedentary time and enhancing sleep quality. 

Response 8: We acknowledge these comments and added a sentence in the Conclusion: “Future behavioural interventions targeting pregnant women should include evidence-based content to improve sleep quality and reduce SED.” (lines 378 - 380).

Reviewer 2 Report

This manuscript does an excellent job analyzing the effects of physical activity intervetions during pregnancy on quality of sleep and sedentarism. The authors found that pregnant women who received structured supervised exercise training had better sleep quality and less sedentary time.

The title is appropriate for the content of the article. The abstract is concise and accurately summarizes the essential information of the paper. The manuscript is written in a clear, direct and active style, free from grammatical errors and other linguistic inconsistencies.

The main strength of this paper is that it describes an actual problem and I appreciate the authors’ patience and professionalism in dealing with this situation. Moreover, this article utilised both subjective and objective methods at different times during pregnancy for data collection.

One possible criticism could be that no sample size nor power calculation was made on the outcomes of the analyses.

I consider that the publication of this article is very important because it opens new perspectives in the field of gynaecology and obstretics. Furthermore, this paper has a high potential due to the new and original information it brings. Based on all these factors, I approve this manuscript.

Author Response

Response to Reviewer 2 Comments

Comments: This manuscript does an excellent job analyzing the effects of physical activity intervetions during pregnancy on quality of sleep and sedentarism. The authors found that pregnant women who received structured supervised exercise training had better sleep quality and less sedentary time.

The title is appropriate for the content of the article. The abstract is concise and accurately summarizes the essential information of the paper. The manuscript is written in a clear, direct and active style, free from grammatical errors and other linguistic inconsistencies.

The main strength of this paper is that it describes an actual problem and I appreciate the authors’ patience and professionalism in dealing with this situation. Moreover, this article utilised both subjective and objective methods at different times during pregnancy for data collection.

One possible criticism could be that no sample size nor power calculation was made on the outcomes of the analyses.

I consider that the publication of this article is very important because it opens new perspectives in the field of gynaecology and obstretics. Furthermore, this paper has a high potential due to the new and original information it brings. Based on all these factors, I approve this manuscript.

Response: We would like to thank the reviewer for the very positive feedback. We appreciate the time you took to read our paper and the approval. We are aware that it is a limitation of the study that the analyses were secondary. We state the secondary nature of the study in the abstract (line 20), the introduction (line 68), and the discussion (line 265). Also, in the section on Strengths and limitations we acknowledge the lack of sample size and power calculations: “An additional limitation of this study is that the analysis is secondary; hence, no sample size nor power calculation was made on the outcomes of the present analyses” (lines 355 and 368).
